
SciPost Phys. Lect. Notes 36 (2022)

# Sterile neutrinos as dark matter candidates

**J. Kopp⋆**

Theoretical Physics Department, CERN, Geneva, Switzerland and
Johannes Gutenberg University Mainz, 55099 Mainz, Germany

⋆ jkopp@cern.ch

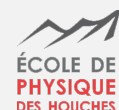

*Part of the Dark Matter*
*Session 118 of the Les Houches School, July 2021*
*published in the Les Houches Lecture Notes Series*

## Abstract

In these brief lecture notes, we introduce sterile neutrinos as dark matter candidates. We discuss in particular their production via oscillations, their radiative decay, as well as possible observational signatures and constraints.

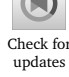
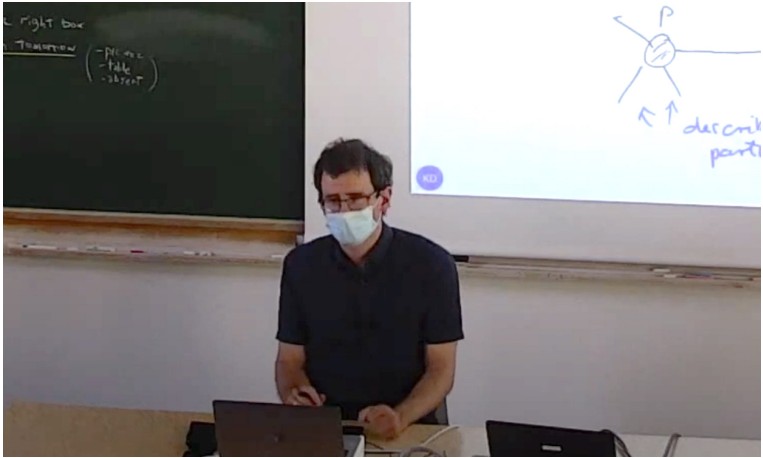

## 1   Neutrino Masses and Mixings

In the simplest extension of the Standard Model (SM) that admits neutrino masses, neutrinos have the following interaction and mass terms:

$$\mathcal{L} = \sum_{\alpha=e,\mu,\tau} \left[ \frac{g}{\sqrt{2}} \left( \overline{\nu_{\alpha,L}} \gamma^\rho e_{\alpha,L} W^+_\rho + h.c. \right) + \frac{g}{2\cos\theta_w} \overline{\nu_{\alpha,L}} \gamma^\rho \nu_{\alpha,L} Z_\rho \right]$$
$$- \sum_{\alpha,\beta=e,\mu,\tau} \left( m_{\alpha\beta} \overline{\nu_{\alpha,L}} \nu_{\beta,R} + h.c. \right). \tag{1}$$

Here, $g$ is the weak coupling constant and $\theta_w$ is the Weinberg angle. Note that only left-handed neutrinos couple to the weak gauge bosons $W^\pm$ and $Z$. In terms of Feynman diagrams, the neutrino interaction vertices can be written as

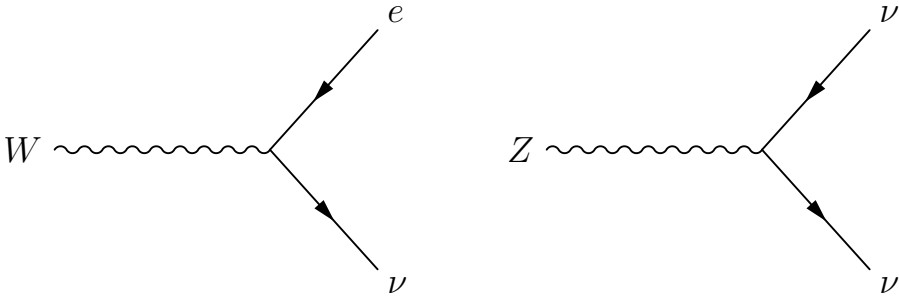

Note that the mass term $\sum_{\alpha,\beta=e,\mu,\tau} m_{\alpha\beta} \overline{\nu_{\alpha,L}} \nu_{\beta,R}$ in eq. (1) is in general *off-diagonal* (i.e. $m_{\alpha\beta}$ can be non-zero even if $\alpha \neq \beta$). This means that the *flavor eigenstates* or *interaction eigenstates* $\nu_\alpha$ ($\alpha = e,\mu,\tau$) do not have a definite mass.

Adding a sterile neutrino $\nu_s$ – i.e. a SM singlet fermion – to this Lagrangian is straightforward. $\nu_s$ appears in the neutrino mass term, but not in the weak interaction term. The mass term then changes into

$$\mathcal{L}_m = - \sum_{\alpha,\beta=e,\mu,\tau,s} \left( m_{\alpha\beta} \overline{\nu_{\alpha,L}} \nu_{\beta,R} + h.c. \right). \tag{2}$$

The mass matrix $m$ can be diagonalized according to

$$m = U m_D V^\dagger, \tag{3}$$

where $m_D = \text{diag}(m_1, m_2, m_3, m_4)$ is a diagonal matrix and $U$, $V$ are unitary matrices. We define the fields in the neutrino *mass eigenstate* basis according to

$$\nu_{j,L} \equiv \sum_\alpha U^*_{\alpha j} \nu_{\alpha,L}, \tag{4}$$

$$\nu_{j,R} \equiv \sum_\alpha V^*_{\alpha j} \nu_{\alpha,R}. \tag{5}$$

In terms of the mass eigenstates, the Lagrangian (1) can be written as

$$\begin{aligned}
\mathcal{L} = &\sum_{\alpha=e,\mu,\tau} \sum_{j=1,2,3,4} \frac{g}{\sqrt{2}} \left( \overline{\nu_{j,L}} U^*_{\alpha j} \gamma^\rho e_{\alpha,L} W^+_\rho + h.c. \right) \\
&+ \sum_{\alpha=e,\mu,\tau} \sum_{j,k=1,2,3,4} \frac{g}{2\cos\theta_w} \overline{\nu_{j,L}} U^*_{\alpha j} \gamma^\rho U_{\alpha k} \nu_{k,L} Z_\rho \\
&- \sum_{j=1,2,3,4} \left( m_j \overline{\nu_{j,L}} \nu_{j,R} + h.c. \right).
\end{aligned} \tag{6}$$

Thus, a charged current neutrino interaction produces a superposition of mass eigenstates, for instance

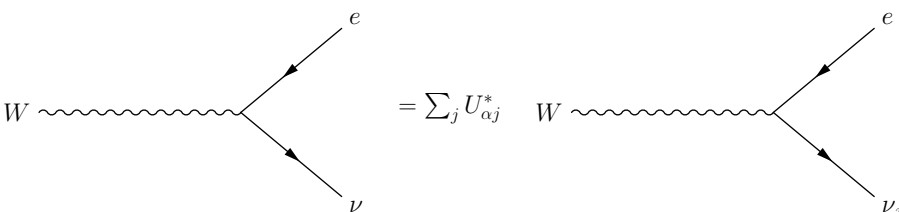

## 2 Neutrino Oscillations

Since neutrino flavor eigenstates—the states that are produced by the weak interaction—are superpositions of mass eigenstates—the states with well-defined kinematics and propagators—we expect quantum interference effects in neutrino experiments. These effects are the neutrino oscillations.

We start from the weak interaction Lagrangian in the mass basis, eq. (6). It implies that the neutrino state of flavor $\alpha$ ($\alpha = e, \mu, \tau, s$) produced in a weak interaction can be written as the following superposition of mass eigenstates

$$|\nu_\alpha\rangle = \sum_j U^*_{\alpha j} |\nu_j\rangle. \tag{7}$$

Note that even though the transformation of the field operators is $\nu_{\alpha,L} = \sum_j U_{\alpha j} \nu_{j,L}$, the transformation of the ket-states is determined by $U^\dagger$ rather than $U$. The reason is that these states are produced by the creation operator $\sum_j U^*_{\alpha j} \bar{\nu}_{j,L}$, not by the annihilation opearators $\sum_j U_{\alpha j} \nu_{j,L}$. Treating $|\nu_j\rangle$ as plane wave states, the wave function at a distance $L$ from the production point, and at a time $T$ after production, is given by

$$|\nu_\alpha(T,L)\rangle = \sum_j U^*_{\alpha j} e^{-iE_j T + ip_j L} |\nu_j\rangle. \tag{8}$$

Note that the energy $E_j$ and the momentum $p_j$ are in general different for the different mass eigenstates because the kinematics of the production process is different for different mass.

A neutrino detector measures the neutrino flavor, i.e. it detects the neutrino in a state

$$\langle \nu_\beta | = \sum_j U_{\beta j} \langle \nu_j |. \tag{9}$$

Therefore, the amplitude for a neutrino produced as $|\nu_\alpha\rangle$ to be detected as $\langle \nu_\beta|$ is

$$\langle \nu_\beta | \nu_\alpha(T,L) \rangle = \sum_{j,k} U_{\alpha j}^* U_{\beta k} e^{-iE_j T + ip_j L} \langle \nu_k | \nu_j \rangle \tag{10}$$

$$= \sum_j U_{\alpha j}^* U_{\beta j} e^{-iE_j T + ip_j L}. \tag{11}$$

The *oscillation probability* is thus

$$P_{\alpha\beta}(T,L) = |\langle \nu_\beta | \nu_\alpha(T,L) \rangle|^2 = \sum_{j,k} U_{\alpha j}^* U_{\beta j} U_{\alpha k} U_{\beta k}^* e^{-i(E_j - E_k)T + i(p_j - p_k)L}. \tag{12}$$

In a typical neutrino oscillation experiment, we do not know when precisely each neutrino is produced (the experimental uncertainty in the production time is much larger than the energy uncertainty of each individual neutrino). Therefore, we should integrate over $T$:[1]

$$P_{\alpha\beta}(L) = \frac{1}{N} \int dT \, P_{\alpha\beta}(T,L) \tag{13}$$

$$= \frac{1}{N} \sum_{j,k} U_{\alpha j}^* U_{\beta j} U_{\alpha k} U_{\beta k}^* \exp\left[ i\left( \sqrt{E^2 - m_j^2} - \sqrt{E^2 - m_k^2} \right)L \right] 2\pi \delta(E_j - E_k) \tag{14}$$

$$\simeq \sum_{j,k} U_{\alpha j}^* U_{\beta j} U_{\alpha k} U_{\beta k}^* \exp\left[ -i \frac{\Delta m_{jk}^2 L}{2E} \right]. \tag{15}$$

Here, $N$ is a normalization constant, which is chosen such that $\sum_\beta P_{\alpha\beta}(L) = 1$. In the last line of eq. (15), we have made the approximation $|m_j^2 - m_k^2| \ll E^2 = E_j^2 = E_k^2$ (equal energy approximation) and carried out a Taylor expansion in the mass squared difference

$$\Delta m_{jk}^2 \equiv m_j^2 - m_k^2. \tag{16}$$

We could also have made the (somewhat unjustified) assumption that all neutrino mass eigenstates are emitted with the same momentum $p$, but different energies. This assumption would have led to the same result, but with phase factor $\exp[-i\Delta m_{jk}^2 T/(2E)]$ instead of $\exp[-i\Delta m_{jk}^2 L/(2E)]$. Since neutrinos travel at the speed of light (up to negligible corrections of order $\Delta m_{jk}^2/E^2$), we can set $L = T$, so that the two approaches become completely equivalent.

The expression for $P_{\alpha\beta}(L)$ becomes particularly simple in the 2-flavor approximation, where the mixing matrix $U$ can be written as

$$U = \begin{pmatrix} \cos\theta & \sin\theta \\ -\sin\theta & \cos\theta \end{pmatrix}. \tag{17}$$

---

[1]This approach may not be strictly valid any more for long-baseline oscillation experiments which typically have excellent timing resolution due to short beam spills and good detector resolution. A more rigorous calculation describing the neutrino as a wave packet confirms, however, that the expression for the oscillation probability remains correct even for long-baseline oscillation experiments.

For instance, if the two flavors are $e$ and $\mu$, we obtain

$$P_{e\mu}^{2\text{-flavor}}(L) = |U_{e1}|^2 |U_{\mu 1}|^2 + |U_{e2}|^2 |U_{\mu 2}|^2$$

$$+ U_{e1} U_{\mu 1} U_{e2} U_{\mu 2} \left[ \exp\left( -i\frac{\Delta m^2 L}{2E} \right) + \exp\left( +i\frac{\Delta m^2 L}{2E} \right) \right] \tag{18}$$

$$= 2\cos^2\theta \sin^2\theta - 2\cos^2\theta \sin^2\theta \cos\left[ \frac{\Delta m^2 L}{2E} \right] \tag{19}$$

$$= \frac{1}{2}\sin^2 2\theta \left( 1 - \cos\left[ \frac{\Delta m^2 L}{2E} \right] \right) \tag{20}$$

$$= \sin^2 2\theta \, \sin^2\left[ \frac{\Delta m^2 L}{4E} \right]. \tag{21}$$

Several comments are in order here:

- States with different energy and momentum $(E_j, p_j)$, $j = 1, 2, 3$ can interfere only if the energy and momentum uncertainties associated with the production and detection processes are larger than $|E_j - E_k|$, $|p_j - p_k|$. This is always satisfied in practice as the typical momentum uncertainty associated with a neutrino production process is at least of the order of an inverse interatomic distance, i.e. of order keV. Therefore, the interference conditions for different neutrino mass eigenstates is easily satisfied.

- A related point: since the energy and momentum uncertainties are so important for interference to happen, it is not correct to treat neutrinos as plane waves. A wave packet formalism is more appropriate.

- The approximation $\sqrt{E^2 - m_j^2} - \sqrt{E^2 - m_k^2} \simeq -\Delta m_{jk}^2/(2E)$ does not require $m_j, m_k \ll E$, but only $m_j^2 - m_k^2 \ll E^2$. (This is sufficient for writing $m_j^2 = m_k^2 + \Delta m_{jk}^2$ and then expanding in $\Delta m_{jk}^2$.)

- For antineutrinos, the above derivation goes through in exactly the same way, except that $U$ should be replaced by $U^*$ everywhere. This is because an antineutrino is created by the field operator $\nu$ rather than the operator $\bar{\nu}$, and the corresponding weak interaction term in the Lagrangian (1) is the hermitian conjugate of the term creating neutrinos. We denote oscillation probabilities for $\bar{\nu}_\alpha \to \bar{\nu}_\beta$ transitions by $P_{\bar{\alpha}\bar{\beta}}(L)$.

## 3 Sterile Neutrinos as Dark Matter Candidates

Sterile neutrinos with masses $>$ keV have all the properties required to account for the DM in the Universe: they are electrically neutral, become non-relativistic early on (thus forming cold dark matter), can have very weak couplings with other particles (if the relevant mixing angles are small), and are stable over cosmological time scales.

An important question for any DM candidate is how the DM abundance observed in the Universe is determined. For the case of sterile neutrinos, the minimal mechanism is the *Dodelson–Widrow* mechanism [1], which we will outline now.

The assumption is that, very early on, no sterile neutrinos exist. Later, they are produced via active-to-sterile ($\nu_a \to \nu_s$) neutrino oscillations. For $\mathcal{O}(\text{keV})$ masses, the oscillation length/oscillation time scale $L^{\text{osc}} = 4\pi E/\Delta m^2$ is very small, so a $\nu_a$–$\nu_s$ superposition is produced very quickly. In the subsequent discussion, is is therefore sufficient to consider the averaged effect of oscillations. For small mixing angle, the neutrino state then consists mostly of $\nu_a$, with a small ($\sim \frac{1}{2}\sin^2 2\theta$) admixture of $\nu_s$. Neutrino collisions with other particles act

as quantum mechanical "measurements", collapsing the wave function either into $\nu_s$ (with a probability of $\frac{1}{2}\sin^2 2\theta$), or into $\nu_a$ (with a probability of $1 - \frac{1}{2}\sin^2 2\theta$). Afterwards, oscillations start again. Active neutrinos again acquire a $\nu_s$ component $\sim \frac{1}{2}\sin^2 2\theta$, and sterile neutrinos acquire a $\nu_a$ component of the same magnitude. However, since $\nu_s$ are much less abundant than $\nu_a$, the back-conversion is negligible. After many collisions, the sterile neutrino abundance has increased to the level observed today. Eventually, collisions cease because the primordial gas becomes too diluted, and the $\nu_s$ abundance present at this time "freezes in". Note that, before freeze-in, active neutrinos are continuously replenished via pair production or charged current interactions.

Dodelson–Widrow production of sterile neutrinos is described by the Boltzmann equation

$$\left(\frac{\partial}{\partial t} - H E \frac{\partial}{\partial E}\right) f_s(E,t) = \left[\frac{1}{2}\sin^2(2\theta_M(E,t))\Gamma(E,t)\right] f_a(E,t),\qquad(22)$$

where $f_s(E,t)$ and $f_a(E,t)$ are the time-dependent momentum distribution functions of sterile and active neutrinos, respectively, and $H$ is the Hubble parameter. Before the interactions between active neutrinos and other SM particles freeze out (the epoch relevant here because the mechanism relies on these collisions), $f_a(E,t)$ is just a Fermi–Dirac distribution

$$f_a(E,t) = \frac{1}{e^{E/T}+1}.\qquad(23)$$

The quantity

$$\Gamma(E,t) \simeq \frac{7\pi}{24}G_F^2 T^4 E \qquad(24)$$

in eq. (22) is the active neutrino interaction rate. The expression in square brackets is thus the probability for the neutrino state to collapse to $\nu_s$ in a collision.[2] $\theta_M(E,t)$ denotes the mixing angle in matter. The second term on the left hand side of eq. (22) describes the change in the energy spectrum due to redshift. Indeed, we have

$$\frac{d}{dt}f_s(E,t) = \left(\frac{\partial}{\partial t} + \frac{dE}{dt}\frac{\partial}{\partial E}\right)f_s(E,t),\qquad(25)$$

and, for relativistic neutrinos, $dE/dt = d(E_0 a^{-1})/dt = -E_0 a^{-2}\dot{a} = -H E$, with $a$ the scale factor of the Universe and $H = \dot{a}/a$. It is justified to assume relativistic neutrinos here as Dodelson–Widrow production peaks at a temperature of order 100 MeV.

From eq. (22), we can compute an evolution equation also for the ratio of number densities of sterile and active neutrinos, $r(t) \equiv n_s(t)/n_a(t)$, with $n_i(t) = 2\int d^3p\, f_i(E,t)/(2\pi)^3$. In doing so, it is convenient to go from derivatives with respect to $t$ to derivatives with respect to $a(t)$. We use

$$\frac{d}{da}n_s(t) = \frac{d}{da}2\int \frac{d^3p}{(2\pi)^3}f_s(E,t)\qquad(26)$$

$$= 2\frac{d}{da}\int \frac{4\pi E^2 dE}{(2\pi)^3}f_s(E,t)\qquad(27)$$

$$= \frac{2}{\dot{a}}\int \frac{4\pi E^2 dE}{(2\pi)^3}\frac{\partial}{\partial t}f_s(E,t) - 2\int \frac{4\pi E^2 dE}{(2\pi)^3}\frac{E}{a}\frac{\partial}{\partial E}f_s(E,t) - 6\int \frac{4\pi E\, dE}{(2\pi)^3}\frac{E}{a}f_s(E,t),$$
$$(28)$$

---

[2]It may seem odd that the neutrino can collapse into $\nu_s$ even though only $\nu_a$ interact. This paradox can only be resolved in a more careful treatment of the Dodelson–Widrow mechanism using the density matrix formalism.

or, equivalently,

$$\dot{a}\frac{d}{da}n_s = 2\int\frac{4\pi E^2\,dE}{(2\pi)^3}\frac{\partial}{\partial t}f_s(E,t) - 2\int\frac{4\pi E^2 dE}{(2\pi)^3}HE\frac{\partial}{\partial E}f_s(E,t) - 3Hn_s. \tag{29}$$

We can thus rewrite eq. (22) as

$$\dot{a}\frac{d}{da}n_s + 3Hn_s = \gamma n_a, \tag{30}$$

where we have defined

$$\gamma \equiv \frac{1}{n_a}\int\frac{d^3p}{(2\pi)^3}\sin^2(2\theta_M)\Gamma(E,t)\frac{1}{e^{p/T}+1}. \tag{31}$$

Since, moreover,

$$\frac{d}{da}n_a = -\frac{3}{a}n_a, \tag{32}$$

we obtain

$$\dot{a}\frac{d}{da}r + \dot{a}\frac{r}{n_a}\frac{d}{da}n_a + 3Hr(t) = \gamma, \tag{33}$$

$$\Leftrightarrow \quad \dot{a}\frac{d}{da}r = \gamma, \tag{34}$$

$$\Leftrightarrow \quad aH\frac{d}{da}r = \gamma, \tag{35}$$

$$\Leftrightarrow \quad \frac{dr}{d\ln a} = \frac{\gamma}{H}. \tag{36}$$

Note that, in the above derivation, we have neglected the time-dependence of the effective number of relativistic degrees of freedom, $g_*$. At epochs where $g_*$ changes, the dependence of energy on the scale factor is no longer simply $E \propto a^{-1}$ because the energy of degrees of freedom that disappear is distributed among those remaining in thermal equilibrium. When this is taken into account, eq. (22) turns into [1]

$$\frac{d}{d\ln a}r = \frac{\gamma}{H} + r\frac{d}{d\ln a}g_*. \tag{37}$$

# 4 Sterile Neutrino Decay

We have mentioned above that any DM candidate should be stable over cosmological time-scales. For sterile neutrinos this is the case in the sense that they live much longer than the age of the Universe if their mixing angles with active neutrinos are sufficiently small. But they are not absolutely stable. A massive, mostly sterile neutrino $\nu_4$ with a small admixture of a light, mostly active neutrino state $\nu_1$ can decay through the following diagrams:

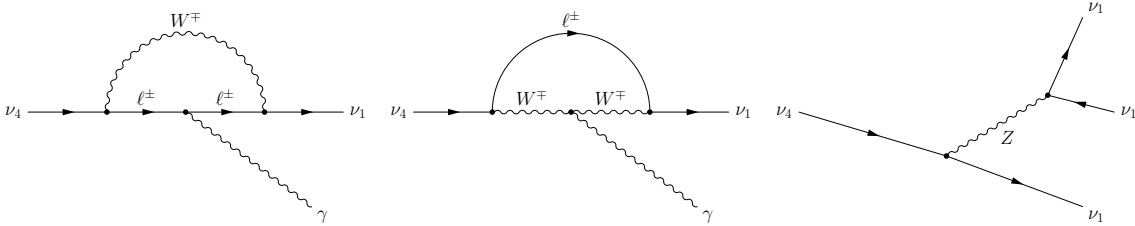

The third of these is phenomenologically irrelevant because the decay products are invisible. It can only be used to impose the constraint that the lifetime of $\nu_4$ should be much larger than the age of the Universe to provide a successful DM candidate. The first two diagrams, on the other hand, lead to radiative neutrino decay $\nu_4 \to \nu_1 \gamma$.

The rate for for $\nu_4 \to \nu_a + \gamma$ is [2–6]

$$
\Gamma(\nu_4 \to \nu_a \gamma)^{\mathrm{D}} = \frac{9\alpha_{\mathrm{em}} G_F^2 m_4^5}{512\pi^4} \sum_{j=1,2,3} \left(1 - \frac{m_j^2}{m_4^2}\right)^3 \left(1 + \frac{m_j^2}{m_4^2}\right) \left| \sum_{\alpha=e,\mu,\tau} \left(1 - \frac{m_\alpha^2}{2M_W^2}\right) U_{\alpha 4} U_{\alpha j}^* \right|^2
$$

$$
\simeq 2.73 \times 10^{-22} \, \mathrm{sec}^{-1} \times \sin^2\theta \times \left(\frac{m_4}{\mathrm{keV}}\right)^5 \tag{38}
$$

in the case of Dirac neutrinos, and [5–7]

$$
\Gamma(\nu_4 \to \nu_a \gamma)^{\mathrm{M}} = \frac{9\alpha_{\mathrm{em}} G_F^2 m_4^5}{256\pi^4} \sum_{j=1,2,3} \left(1 - \frac{m_j^2}{m_4^2}\right)^3 \left\{ \left(1 + \frac{m_j^2}{m_4^2}\right)^2 \left[ \sum_{\alpha=e,\mu,\tau} \left(1 - \frac{m_\alpha^2}{2M_W^2}\right) \mathrm{Im}(U_{\alpha 4} U_{\alpha j}^*) \right]^2 \right.
$$

$$
\left. + \left(1 - \frac{m_j^2}{m_4^2}\right)^2 \left[ \sum_{\alpha=e,\mu,\tau} \left(1 - \frac{m_\alpha^2}{2M_W^2}\right) \mathrm{Re}(U_{\alpha 4} U_{\alpha j}^*) \right]^2 \right\}
$$

$$
\simeq 5.46 \times 10^{-22} \, \mathrm{sec}^{-1} \times \sin^2\theta \times \left(\frac{m_4}{\mathrm{keV}}\right)^5 \tag{39}
$$

for Majorana neutrinos. In these expressions, $m_j$ ($j = 1..4$) are the neutrino mass eigenvalues, $m_e$, $m_\mu$, and $m_\tau$ denote the charged lepton masses, $M_W$ is the $W$ boson mass, $G_F$ is the Fermi constant, and $\alpha_{\mathrm{em}}$ is the electromagnetic fine structure constant. The numerical approximations in eqs. (38) and (39) were obtained in the limit $m_j \ll m_4$ and $m_\alpha \ll M_W$. Moreover, in this limit, the dependence on the mixing matrix elements can be expressed in terms of the effective mixing angle $\sin^2\theta \equiv \sum_j |U_{s4} U_{sj}^*|^2$. Assuming that $\nu_4$ mixes predominantly with only one of the light mass eigenstates, and that the corresponding mixing angle is $\ll 1$, $\theta$ can be identified with that mixing angle. The fact that the expression is different for the two cases comes from the fact that, for Dirac neutrinos, only an $\ell^-$ and a $W^+$ can propagate in the loop (opposite for Dirac antineutrinos), while for Majorana neutrinos, also the combination $\ell^+$ and $W^-$ is possible.

The radiative decay mode implies that, in spite of its small rate (much smaller than the inverse age of the Universe), sterile neutrino DM leads to potentially observable, nearly monoenergetic $\mathcal{O}(\mathrm{keV})$ X-ray emission in regions of high DM density (Galactic Center, galaxy clusters, etc.). We call the emission "nearly monoenergetic" because the DM velocity dispersion induces Doppler broadening. For DM in a galaxy cluster, with a typical velocity dispersion of order $v \sim 1000 \, \mathrm{km/sec}$, the relative line width is $\sqrt{(1+v)/(1-v)} - 1 \sim 0.3\%$. Searches for mono-energetic X-rays have been carried out, and results will be discussed below.

## 5 Constraints on Sterile Neutrino Dark Matter

We have seen in eqs. (38) and (39) that the decay rate of sterile neutrino DM is tiny. However, a modern X-ray telescope sees about $10^{78}$ dark matter particles in its line of sight to a nearby galaxy cluster of mass $10^{15} \, \mathrm{M}_\odot$, so a signal may be detectable.

The resulting constraints on sterile neutrino dark matter are summarized in fig. 1. We see that only mixing angles as small as $\sin^2 2\theta \lesssim 10^{-11}$ are still allowed. Even for these, the

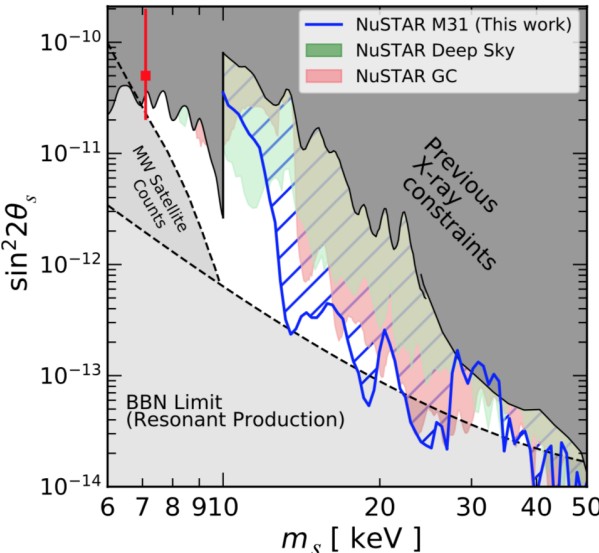

Figure 1: Constraints on sterile neutrino dark matter. Figure taken from [8]. The colored and gray regions show limits from X-ray searches, while the medium gray region on the left labeled "MW Satellite Counts" is based on structure formation arguments [9]. The region labeled "BBN Limit (Resonant Production)" is disfavored by BBN constraints on the lepton asymmetry if the latter is invoked to enhance sterile neutrino production. The red dot with an error bar indicates the parameters corresponding to the sterile neutrino explanation of the 3.5 keV line [10, 11].

parameter space is very limited. It is also important to note that, for such small mixing angles, the Dodelson–Widrow mechanism can no longer produce the observed DM abundance. There are alternative mechanisms, though, that can populate this region of parameter space, including for instance production in the decay of heavy particles, or production through resonant oscillations. The latter mechanism, called the Shi–Fuller mechanism [12], assumes that the lepton asymetry of the Universe is sizeable (much larger than the baryon asymmetry); in this case, neutrinos feel an extra Mikheyev–Smirnov–Wolfenstein (MSW) potential which can enhance the effective mixing angle in the early Universe, while keeping the vacuum mixing angle relevant to observations today small. The problem is that large lepton asymmetries are difficult to achieve in baryogenesis/leptogenesis models, and they are moreover constrained by BBN. This constraint is shown in light gray at the bottom of fig. 1. Because smaller mixing angles require larger lepton asymmetries in the Shi–Fuller mechanism, the BBN bound is actually a *lower* bound on the mixing angle.

A further constraint arises from large-scale structure formation. Namely, DM particles with $\mathcal{O}(\text{keV})$ masses are still moving relatively fast at the time when structure formation starts (matter–radiation equality, $T \sim \text{eV}$). They can therefore wash out small-scale density inhomogeneities, and this affects the subsequent formation of galaxies. In particular, small structures like dwarf galaxies are then less likely to be produced. Based on the observed counts of dwarf galaxies accompanying the Milky Way, one obtains the constraint shown in medium gray in the left part of fig. 1. Note that the constraint as shown here is only valid for sterile neutrinos produced via the Shi–Fuller mechanism. A similar limit could also be derived for other production mechanisms, but because each production mechanism leads to a different energy distribution for the sterile neutrinos at production, the constraint depends on the production mechanism.

Sterile neutrino dark matter is also constrained by phase space arguments: if its mass was

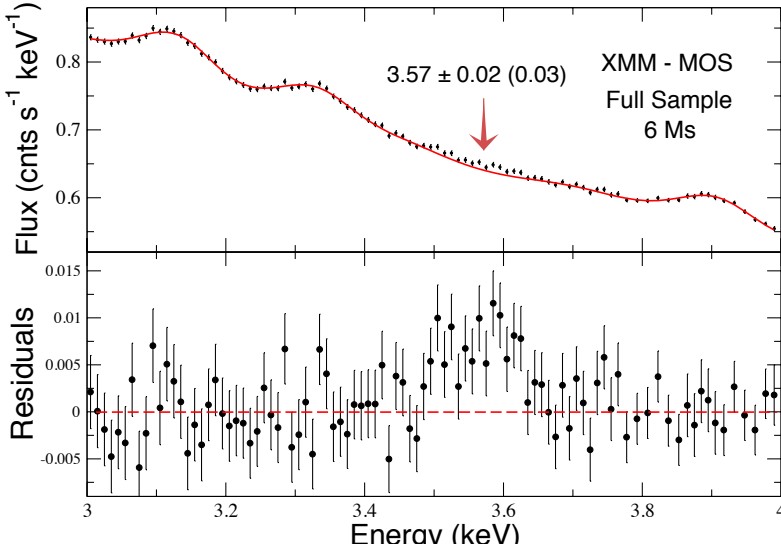

Figure 2: Stacked XMM-Newton spectra from a number of galaxy clusters, leading to the detection of an unexplained feature at around 3.5 keV. Plot taken from [10].

too low ($\lesssim 1$ keV), the conservation of phase space density would forbid the formation of compact galactic cores. This constraint is called the Tremaine–Gunn bound [13]. A slightly weaker bound arises also from the Pauli exclusion principle, which prohibits arbitrarily dense packing of fermionic dark matter, again in conflict with observations of galactic cores. Finally, observations of cosmic structure on relatively small scales ($\lesssim 10$ Mpc) using Lyman-$\alpha$ forests constrain keV-scale dark matter [14], though the quantitative power of these constraint depends on the dark matter production mechanism.

## 6 The 3.5 keV Anomaly

In 2014, stacked observations of galaxy clusters using data from the XMM-Newton X-ray telescope have led to the detection of an unidentified X-ray line near 3.55 keV [10], see fig. 2. (The width of the excess is compatible with the instrumental resolution, so calling the excess a line is justified.)

The possibility that this line constitutes a detection of the radiative decay of a $\sim 7$ keV sterile neutrino has caused a lot of excitement. Comparing the observed flux from different astrophysical objects (galaxies, galaxy clusters, . . . ), the scaling with the DM abundance in these objects is not as perfect as one would hope, but also not bad enough to definitely rule out new physics as an explanation, see for instance [15]. There is a heated debate going on about the trustworthiness of both the positive observations and the null results. Fortunately, future X-ray telescopes with higher energy resolution should be able to dicriminate between a dark matter origin of the signal and more mundane explanation in terms of atomic physics effects. This will be possible because the precise shape of the line is predicted to be different in the two cases.

# Acknowledgements

It is a great pleasure to thank the organizers of the Les Houches Summer School 2021 on Dark Matter, the students, and the local staff for making this event a success.

**Further reading.** While we hope that these brief lecture notes offer a compact introduction to the subject of sterile neutrinos as dark matter candidates, they necessarily cannot be fully comprehensive. For instance, we did not comment on the large number of models that exist beyond the minimal Dodelson–Widrow and Shi–Fuller scenarios. A much more detailed discussion of this and other aspects can be found in numerous excellent reviews in the literature, see for instance refs. [16–20].

**Funding information.** The author's work has been partially supported by the European Research Council (ERC) under the European Union's Horizon 2020 research and innovation program (grant agreement No. 637506, "$\nu$Directions"). He has also received funding from the German Research Foundation (DFG) under grant No. KO 4820/4-1.

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
