# Peer review of "Sterile Neutrinos as Dark Matter Candidates"

_SciPost Physics Lecture Notes, doi:SciPost Phys. Lect. Notes 36 (2022)_

## Round 1 · Referee Report · Anonymous (Referee 1) · 2021-9-26

Report
I would like to suggest a small number of minor editorial corrections, listed I the pdf file attached.
Author: Joachim Kopp on 2021-10-12 [id 1839]
(in reply to Report 1 on 2021-09-26)
I would like to thank the referee for their careful reading and detailed comments / suggestions. They will all be incorporated in the next version of the manuscript, with the exception of the very first one:
'Page 4, second line after Eq (16): “This would have led ...” should be replaced with “An alternative assumption X would have led ...”, where X is an assumption different from the one mentioned in the previous line.'
The sentence is correct as it is - the statement about the modified phase factor refers to the (somewhat unjustified) equal-momentum assumption, not to yet another alternative assumption.
I attach a PDF that highlights the changes made in response to this referee report, as well as to a list of very detailed and useful comments submitted by Pueh-Leng Tan, one of the students who attended the lectures at the Les Houches School.
Author: Joachim Kopp on 2021-10-26 [id 1880]
(in reply to Report 3 on 2021-10-22)Thank you for these useful comments. I've addressed most of them (mostly by appropriate rewording), and I reply here explicitly to some of them:
I attach once again a PDF file that highlights the changes made in response to this and the other referee reports.
Attachment:
diff_b2BpZQr.pdf

---

## Round 1 · Referee Report · Anonymous (Referee 2) · 2021-10-14

Report
This evaluation and comments are based on the updated version uploaded by the author to this website on October 12, 2021.
Requested changes
Please see list in the PDF attached.
Author: Joachim Kopp on 2021-10-26 [id 1879]
(in reply to Report 2 on 2021-10-14)I wold like to thank the referee for their careful reading of the manuscript and their useful and detailed comments. I've taken them into account in v2, and I attach a file that highlights the changes made in response to this (and the other) referee reports compared to v1.
Attachment:

---

## Round 1 · Referee Report · Anonymous (Referee 3) · 2021-10-22

Strengths
Weaknesses
no specific models named
notation sometimes unclear (chriral vs mass vs interaction eigenstates)
no literature for "further reading" provided for the students (textbook, review or similar)
Report
Before (1), I would reject the statement that "neutrinos have the following interactions and mass term" in the SM. Neutrinos are massless in the SM, and the term shown there is only one possible way to extend the SM to accommodate neutrino masses. Alternatively one could have a Majorana mass term. This possibility should be mentioned.
I find the discussion between (3) and (6) a bit confusing. The labels L and R suggest that the spinors \nu_L and \nu_R are chiral two-component objects, but then it is said that they are "mass eigenstates". This is confusing in two ways. Firstly, spinors are not states, as the author later points out after (7). Secondly, chirality is not conserved for massive fermions, therefore eigenstates of the chiral projectors cannot be mass eigenstates.
If one uses 4-vector notation, the spinors associated with mass eigenstates are usually expressed as Dirac or Majorana spinors. I suppose the author assumes Dirac neutrinos here, and \nu_L and \nu_R are the chiral projections of the Dirac spinors?
I am under the impression that the convention of $U$ and $U^*$ in (7) is not consistent with that in (4).
The quantity E in (14) is not defined. Later it is set equal to $E_i$ and $E_j$ - shouldn't it be either of those to begin with?
I am not sure if I agree with the statement in the beginning of section 3 that the Dodelson-Widrow is the "leading mechanism" for sterile neutrino DM production. It is the only mechanism that works without involving new interactions or large lepton asymmetries, so one may call it "minimal" if one wants. But I do not think that it is "leading" because it cannot actually explain the observed DM abundance (if one takes structure formation bounds seriously).
In the qualitative discussion before (22) the author suggests that sterile neutrinos are produced via oscillations, but (22) cannot actually describe oscillations. For that one would need a density matrix equation.
What (22) describes is the production of a heavy mass eigenstate through its weak interactions. I think a useful way to explain this is to say that the mass eigenstate is actually not sterile, it has a mixing-suppressed weak interaction and can therefore be produced in any weak process (provided that there is enough energy), without involving any oscillations. This viewpoint also makes the production rate in square brackets intuitive, as sin\theta G_F acts as an effective coupling constant. I am sure the author knows all of this. But from a didactic viewpoint I think that it is not ideal to talk about oscillations and then use a formula that cannot describe oscillations.
In many equations following (25) the author treats the neutrinos kinematically as massless ($E=p$), e.g. when suggesting that E gets redshiftet. This is not a bad approximation for keV mass neutrinos because the DW production peaks at 100 MeV or so, but from a didactic viewpoint it may be useful to point this out. Overall I find the discussion of the Boltzmann equation between (22) and (27) a little complicated, e.g. in comparison to the treatment in the textbook of Kolb and Turner.
I personally find the discussion around (38) and (39) a bit confusing because the author uses an interaction eigenstate in the initial state, while it is more common to associate the legs of Feynman diagrams with mass eigenstates ("asymptotic states"). I would say that the long-lived particle that forms the DM is the heavy mass eigenstate... but different people may prefer different viewpoints here.
In figure 1 I would use a plot from some review instead of the NuStar figure, just because I recall that the NuStar bounds were a bit controversial when they came out. I do not remember what the criticism was, so I cannot really say if the plot is correct or not, but I think that it is better to be conservative when one reviews the literature.
I think that it would be helpful to include a more systematic discussion of the different production mechanisms. Apart from Dodelson-Widrow and Shi-Fuller, there are decays of heavy particles (which the author mentions somewhere in the text in passing), and also the possibility that the sterile neutrinos have new gauge interactions at high energies (e.g. in left-right symmetric models). One could list all these possibilities instead of mentioning them here and there in passing.
In section 5 I believe that the Tremaine-Gunnn bound and constraints from Lyman alpha forest observations deserve mentioning.
For the 3.5 keV line, it might be a good idea to mention that future x-ray telescopes with high spectral resolution can help clarify its origin.
I think that it would be helpful to name a few specific models in which sterile neutrino DM can appear. One example would be the type I seesaw, which is probably the most studied scenario.
If this were a research article, I would probably be a bit more insising when it comes to citing original work in a representative manner. In this type of document the lack of original citations is probably acceptable.
Finally, I think that it might be a good idea to recommend some reviews or textbooks for those students who are interested in further reading.

---

## Round 2 · List of Changes

the changes to the manuscript are detailed in the diff file I submitted resubmitted in response to the report from Referee #3

---

## Editorial Decision

published